# Counterfactual Prediction for Bundle Treatment

**Hao Zou[1], Peng Cui[1], Bo Li[1], Zheyan Shen[1], Jianxin Ma[2], Hongxia Yang[2], Yue He[1]**
[1]Tsinghua University, [2]Alibaba Group
zouh18@mails.tsinghua.edu.cn,cuip@tsinghua.edu.cn,libo@sem.tsinghua.edu.cn,
shenzy17@mails.tsinghua.edu.cn,majx13fromthu@gmail.com,
yang.yhx@alibaba-inc.com,heyue18@mails.tsinghua.edu.cn

## Abstract

Estimating counterfactual outcome of different treatments from observational data is an important problem to assist decision making in a variety of fields. Among the various forms of treatment specification, bundle treatment has been widely adopted in many scenarios, such as recommendation systems and online marketing. The bundle treatment usually can be abstracted as a high dimensional binary vector, which makes it more challenging for researchers to remove the confounding bias in observational data. In this work, we assume the existence of low dimensional latent structure underlying bundle treatment. Via the learned latent representations of treatments, we propose a novel variational sample re-weighting (VSR) method to eliminate confounding bias by decorrelating the treatments and confounders. Finally, we conduct extensive experiments to demonstrate that the predictive model trained on this re-weighted dataset can achieve more accurate counterfactual outcome prediction.

## 1   Introduction

Accurately predicting the counterfactual outcome of different treatments is of paramount importance for decision makers across many domains, such as healthcare [4] and marketing [5]. As the prediction tasks being different, the treatment specification can also be in various forms, for instance, binary, continuous, multi-level and bundle. Among these forms, bundle treatment, which can be abstracted as a high dimensional binary vector, has been widely used in many real scenarios. For example, in recommendation systems, the treatment can be a bundle of exposed items which are selected from a large item pool. Each dimension of the treatment means whether the corresponding item is in the bundle.

The availability of ample observational data reveals a promising possibility of applying machine learning to predict the outcome when assigned different treatments. Nevertheless, a nonnegligible challenge in this way is that when we collect observational data, the treatment is usually assigned not randomly, but to be correlated with the confounders which can also affect outcomes. For instance, the exposure of products in recommendation systems is usually been confounded with the profiles and visiting history of the users. Such phenomenon, namely confounding bias, induce the distribution discrepancy between the distribution from which the observational data is collected and the one where the treatment is randomly assigned. This discrepancy may cause the predictive model to focus on the outcome estimation for the treatment that the assignment policy puts larger probability on but ignore the other treatments. Therefore, to obtain the reliable estimation regardless the dependency between treatments and confounders, we need to reduce the confounding bias of observational data by decorrelating confounders and treatments.

There has been a large amount of literature that attempted to resolve the challenge in counterfactual prediction problem. For binary treatment, Johansson et al. [16] introduced the treatment invariant representation learning borrowed from domain adaptation [8, 33, 6] to address the confounding bias.

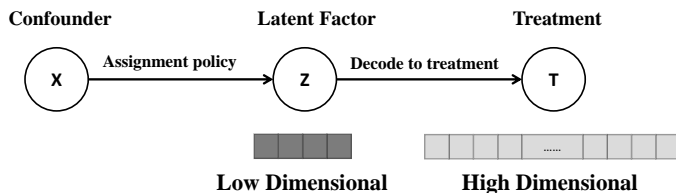

Figure 1: Diagram of generation of bundle treatments.

Based on the treatment invariant representation, Hassanpour and Greiner [11] used the propensity score based sample re-weighting to remove confounding bias more completely. Further, Liu et al. [21] extended the treatment invariant representation to the multi-level treatment setting. Yoon et al. [35] introduced the generative adversarial networks (GAN) to predict counterfactual outcome of multi-level treatment. However, the literature above all focus on single treatment setting, rather than bundle treatment. Although we can naively transform the bundle treatment into multi-level setting by binary coding, the high dimensional property of original treatment may lead to exponential growth on treatment levels which induces infeasible computation complexity for conventional methods.

To address the high-dimension problem of bundle treatment, we assume that there exists a low dimensional latent structure which could generate the original treatment. For example, in recommendation systems, the recommended items in a bundle are usually determined by several latent factors, such as item categories and prices. The data generation process under the bundle treatment setting can be abstracted as the diagram in Figure 1. Therefore, instead of decorrelating the high dimensional original treatments and confounders, it is a natural idea that we can remove the confounding bias through inferring the latent factors of treatments and decorrelating them with confounders.

In this paper, we propose the variational sample re-weighting (VSR) algorithm to decorrelate the confounders and bundle treatments by utilizing the latent structure of treatments. We leverage variational autoencoder (VAE) [18, 26], a widely used generative model to learn the latent representations of treatments, and conduct density ratio estimation [30, 32, 1] based on deep neural networks to decorrelate the confounders and latent representations. Instead of point-wise estimation, we calculate sample weights distribution-wisely by aggregating the density ratio in the whole variational distribution of latent variables. After re-weighting samples by these weights, we can create a pseudo-population where the confounding bias is effectively reduced, with which the off-the-shelf predictive model can achieve more accurate counterfactual outcome estimation. Finally, we conduct extensive experiments on both synthetic datasets and real world datasets to demonstrate the advantages of our proposed variational sample re-weighting algorithm.

## 2 Related Works

**Counterfactual Prediction** Under the binary treatment setting, some literature [16, 29, 34, 11] learned treatment invariant representation of confounders to remove confounding bias, adjust observational distribution and predict counterfactual outcome. As an alternative method, Johansson et al. [17] applied sample re-weighting instead of treatment invariant representation learning. Furthermore, Hassanpour and Greiner [12] assumed that some part of confounders have no effect on treatment assignment or outcome, and proposed to neglect these confounders when learning sample weights. As for multi-level treatment, Yoon et al. [35] introduced GAN and proposed GANITE to predict counterfactual outcome of each treatment. When the existence of unobserved confounders may mislead the counterfactual prediction, Louizos et al. [23] attempt to infer the unobserved confounders from proxies, and Hartford et al. [10] introduced instrumental variable to address this problem.

**Sample Re-weighting in Causal Inference** The traditional methods in causal inference usually re-weight samples based on propensity score [28, 20, 3] to remove confounding bias for binary treatments. Some researchers studied the problem under a multi-level treatment setting by extending the propensity score to generalized propensity score [14, 15, 22]. However, the (generalized) propensity score estimation usually requires correct model specification. This may not be accessible in many applications. To reduce the model dependency problem, some researchers proposed new technology that can directly learn the sample weights by balancing the moments of confounders

[19, 2, 9, 36]. The motivation behind these methods is that the distribution can be determined by the collection of all moments. However, only finite moments can be involved in balancing computation, and this needs to be designed manually by researchers.

# 3 Problem Statement and Approach

In this section, we first introduce the concepts and notations of the problem. Then we detailly give a description of our variational sample re-weighting algorithm.

## 3.1 Counterfactual Prediction for Bundle Treatment

We aim to learn the outcome of each treatment on the individual with different contexts from the observational data. The observational data comprises the confounders $\mathbf{X} \in \mathcal{X} \subset \mathbb{R}^d$, the bundle treatment $\mathbf{T} \in \mathcal{T} \subset \{0,1\}^p$ assigned to the individual and the outcome $\mathbf{y} \in \mathbb{R}$ given the treatment and the confounders. Therefore, the observational data can be notated as $\{(\mathbf{x}_i, \mathbf{t}_i, y_i)\}_{1 \le i \le n}$, where $n$ is the number of samples. We take item recommendation as an example for explanation. The confounder vector $\mathbf{x}_i$ encodes the context information of the user including the user attributes and visiting history, which can influence both the treatment assignment (i.e. item bundle) and outcome (i.e. user's response). The treatment vector $\mathbf{t}_i$ represents the displayed items where each bit $t_{i,j}$ corresponds to an item in the pool. That is $t_{i,j} = 1$ or $0$ means the $j^{th}$ item is displayed or not. The outcome variable $y_i$ means the user's response to the item bundle, such as the click rate.

With the observational data, we hope to learn a hypothesis $f_{\theta_p} : \mathcal{X} \times \mathcal{T} \mapsto \mathbb{R}$ with model parameters $\theta_p$, which predicts the outcome based on the confounder and treatment variables. Counterfactual prediction aims that the learned hypothesis can predict accurate outcome of all the available treatments for each individual. Formally, the prediction error for the individual $\mathbf{X}$ can be defined as $\mathcal{E}(\mathbf{X}) = \mathbb{E}_{\mathbf{T} \sim p(\mathbf{T})}[\mathcal{L}(f_{\theta_p}(\mathbf{X}, \mathbf{T}), \mathbf{y}(\mathbf{X}, \mathbf{T}))]$, where $\mathcal{L}(\cdot, \cdot)$ is the error function (e.g. square error), $\mathbf{y}(\cdot, \cdot)$ is the ground truth of outcome given $\mathbf{X}$ and $\mathbf{T}$. Therefore, the target of counterfactual prediction is minimizing $\mathcal{E}_{cf} = \mathbb{E}_{\mathbf{X} \sim p(\mathbf{X})}[\mathcal{E}(\mathbf{X})]$.

Classical supervised learning can be applied to learn the hypothesis. However, in the observational data, the treatment $\mathbf{T}$ is assigned based on confounders. Directly using supervised machine learning may lead to accurate outcome estimation for the treatment that the assignment policy put larger probability on $p(\mathbf{T}|\mathbf{X})$ but inaccurate outcome estimation for other treatments. To address this distribution discrepancy induced by confounding bias, we assume the standard assumptions [28] in causal inference, such as unconfoundedness, overlap and stable unit treatment value are satisfied. We also introduce the ideas of re-weighting samples $w_i = W_T(\mathbf{x}_i, \mathbf{t}_i)$ from causal inference to decorrelate confounders and treatments for removing confounding bias, and then optimize the prediction error on the re-weighted data $\mathcal{E}_f^w = \mathbb{E}_{\mathbf{X}, \mathbf{T} \sim p(\mathbf{X}, \mathbf{T})}[\mathcal{L}(f_{\theta_p}(\mathbf{X}, \mathbf{T}), \mathbf{y}(\mathbf{X}, \mathbf{T}))W_T(\mathbf{X}, \mathbf{T})]$.

Here, we give an upper bound of $\mathcal{E}_{cf}$ based on $\mathcal{E}_f^w$ and Integral Probability Metric (IPM) [31, 29], which is a metric of distribution distance. Specifically, for two distribution on $\mathcal{X} \times \mathcal{T}$, $p_1(\mathbf{X}, \mathbf{T})$, $p_2(\mathbf{X}, \mathbf{T})$ and a family $G$ of functions $g : \mathcal{X} \times \mathcal{T} \mapsto \mathbb{R}$, we have: $\text{IPM}_G(p_1(\mathbf{X}, \mathbf{T}), p_2(\mathbf{X}, \mathbf{T})) = \sup_{g \in G} |\int_{\mathbf{X}} \int_{\mathbf{T}} (p_1(\mathbf{X}, \mathbf{T}) - p_2(\mathbf{X}, \mathbf{T}))g(\mathbf{X}, \mathbf{T})d\mathbf{X}d\mathbf{T}|$.

**Theorem 1.** *Assuming a family $G$ of functions $g : \mathcal{X} \times \mathcal{T} \mapsto \mathbb{R}$, $l(\mathbf{X}, \mathbf{T}) = \mathcal{L}(f(\mathbf{X}, \mathbf{T}), \mathbf{y}(\mathbf{X}, \mathbf{T})) \in G$ and overlap assumption is satisfied, formally, $p(\mathbf{T}|\mathbf{X}) > 0, \forall \mathbf{T} \in \mathcal{T}, \mathbf{X} \in \mathcal{X}$, we have:*

$$\mathcal{E}_{cf} \le \mathcal{E}_f^w + \text{IPM}_G(W_T(\mathbf{X}, \mathbf{T})p(\mathbf{X}, \mathbf{T}), p(\mathbf{X})p(\mathbf{T})). \tag{1}$$

*More specifically, $\text{IPM}_G = 0$ and $\mathcal{E}_{cf} = \mathcal{E}_f^w$, when $W_T(\mathbf{X}, \mathbf{T}) = p(\mathbf{T})/p(\mathbf{T}|\mathbf{X})$.*

Proof can be found in supplementry material. From Eq. (1), we can observe that if we do not decorrelate treatments and confounders after sample re-weighting, which means $W_T(\mathbf{X}, \mathbf{T})p(\mathbf{X}, \mathbf{T}) \ne p(\mathbf{X})p(\mathbf{T})$, $\text{IPM}_G$ value may be large and directly minimizing $\mathcal{E}_f^w$ can not guarantee the result of $\mathcal{E}_{cf}$. On the other hand, if the confounding bias is successfully removed (e.g. $\text{IPM}_G \approx 0$), we can optimize $\mathcal{E}_{cf}$ by minimizing $\mathcal{E}_f^w$. Under mild assumption, it can be proved that the weights $\mathbf{w}^d = \{w_i^d\}_{i=1}^n = \{p(\mathbf{t}_i)/p(\mathbf{t}_i|\mathbf{x}_i)\}_{i=1}^n$ is one of the optimal sample weights that induce lowest counterfactual prediction error $\mathcal{E}_{cf}$ by minimizing the empirical re-weighted prediction error $\mathcal{E}_f^w$.

However, the high dimensional property of bundle treatment bring challenges to decorrelating confounders and treatments. To address it, we assume that the bundle treatment $\mathbf{T}$ is determined by the

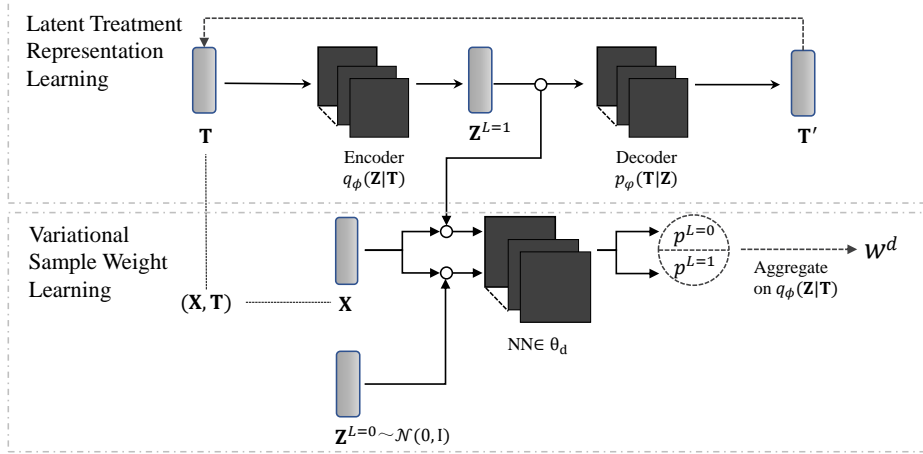

Figure 2: The framework of variational sample re-weighting algorithm.

low dimensional latent representation $\mathbf{Z}$. From Figure 1, we can observe that if the latent representation is decorrelated with confounder in the re-weighted distribution, which means $p_w(\mathbf{Z}|\mathbf{X}) = p_w(\mathbf{Z})$, the treatment is also decorrelated with confounder and the confounding bias is removed. Formally,

$$p_w(\mathbf{T}|\mathbf{X}) = \int_{\mathbf{Z}} p_w(\mathbf{T}|\mathbf{Z})p_w(\mathbf{Z}|\mathbf{X})d\mathbf{Z} = \int_{\mathbf{Z}} p_w(\mathbf{T}|\mathbf{Z})p_w(\mathbf{Z})d\mathbf{Z} = p_w(\mathbf{T}). \tag{2}$$

Therefore, we propose variational sample re-weighting to learn latent representations of treatments and remove confounding bias through decorrelating the latent treatment representations and confounders.

## 3.2 Variational Sample Re-weighting

In this section, we introduce our VSR algorithm. This method can be a data pre-processing method to improve performance of the trained counterfacutal predictive model $f_{\theta_p} : \mathcal{X} \times \mathcal{T} \mapsto \mathbb{R}$.

### 3.2.1 Latent Treatment Representation Learning

To learn the latent representation of treatment, we introduce variational autoencoder (VAE), which makes weaker assumption on the data generation process and latent structure. It simultaneously learns encoder $q_\phi(\mathbf{Z}|\mathbf{T})$ and decoder $p_\varphi(\mathbf{T}|\mathbf{Z})$ by maximizing evidence lower bound (ELBO):

$$\mathcal{L}_{ELBO} = \frac{1}{n} \sum_{i=1}^{n} \mathbb{E}_{\mathbf{z} \sim q_\phi(\mathbf{z}|\mathbf{t}_i)}[\log p_\varphi(\mathbf{t}_i|\mathbf{z}) + \log p(\mathbf{z}) - \log q_\phi(\mathbf{z}|\mathbf{t}_i)], \tag{3}$$

where $p(\mathbf{Z})$ is the prior distribution of the latent treatment representation and usually assumed to be an standard normal distribution $\mathcal{N}(0, \mathbf{I})$. With the encoder component, we can infer the latent representation $\mathbf{z} \sim q_\phi(\mathbf{z}|\mathbf{t}_i)$ of treatment $\mathbf{t}_i$. Then we transform the original dataset to the joint space of confounders and latent representations, that is $\{(\mathbf{x}_i, \mathbf{z})\}_{1 \leq i \leq n}$, $\mathbf{z} \sim q_\phi(\mathbf{z}|\mathbf{t}_i)$. In the target decorrelated distribution, the latent representations should be assigned to each sample independently of confounders, formally, the target distribution is $\{(\mathbf{x}_i, \mathbf{z})\}_{1 \leq i \leq n}$, $\mathbf{z} \sim p(\mathbf{z})$.

### 3.2.2 Variational Sample Weight Learning

The classical sample weights learning methods in causal inference field almostly focused on the single treatment setting, which can hardly be extended to decorrelate the latent treatment representation and confounders. To address this problem, we introduce density ratio estimation based on deep neural networks [30]. Since the learned latent representation of treatment is of a variational distribution $q_\phi(\mathbf{z}|\mathbf{t}_i)$ rather than a single point, we can not naively use point-wise estimation of density ratio in latent space as the sample weight. Instead, considering all the point-wise estimation of density ratio in the variational distribution, we start from $W_T(\mathbf{X}, \mathbf{T}) = p(\mathbf{T})/p(\mathbf{T}|\mathbf{X})$ [7, 27] and propose variational sample weight which is calculated distribution-wisely as following:

$$w_i^d = W_T(\mathbf{x}_i, \mathbf{t}_i) = \frac{p(\mathbf{t}_i)}{p(\mathbf{t}_i|\mathbf{x}_i)} = \frac{p(\mathbf{t}_i)}{\int_{\mathbf{z}} p(\mathbf{z}|\mathbf{x}_i)p(\mathbf{t}_i|\mathbf{z})d\mathbf{z}} = \frac{1}{\int_{\mathbf{z}} p(\mathbf{z}|\mathbf{x}_i)\frac{p(\mathbf{t}_i|\mathbf{z})}{p(\mathbf{t}_i)}d\mathbf{z}}$$

$$= \frac{1}{\int_{\mathbf{z}} p(\mathbf{z}|\mathbf{x}_i)\frac{p(\mathbf{z}|\mathbf{t}_i)}{p(\mathbf{z})}d\mathbf{z}} = \frac{1}{\int_{\mathbf{z}} p(\mathbf{z}|\mathbf{t}_i)\frac{p(\mathbf{z}|\mathbf{x}_i)}{p(\mathbf{z})}d\mathbf{z}} = \frac{1}{\int_{\mathbf{z}} p(\mathbf{z}|\mathbf{t}_i)\frac{p(\mathbf{z},\mathbf{x}_i)}{p(\mathbf{z})p(\mathbf{x}_i)}d\mathbf{z}}$$

$$= \frac{1}{\int_{\mathbf{z}} p(\mathbf{z}|\mathbf{t}_i)\frac{1}{W_Z(\mathbf{x}_i,\mathbf{z})}d\mathbf{z}} = \frac{1}{\mathbb{E}_{\mathbf{z}\sim q_\phi(\mathbf{z}|\mathbf{t}_i)}\left[\frac{1}{W_Z(\mathbf{x}_i,\mathbf{z})}\right]}, \tag{4}$$

where $W_Z(\mathbf{X}, \mathbf{Z})$ is the density ratio estimation for the points in space $\mathcal{X} \times \mathcal{Z}$ to decorrelate $\mathbf{X}$ and $\mathbf{Z}$. It can be guaranteed that with the weights $\mathbf{w}^d = \{w_i^d\}_{i=1}^n$, the treatments can be decorrelated with confounders in the observational data, and therefore we can obtain a pseudo-data with alleviated confounding bias. Besides, as the variational sample weights are estimated distribution-wisely from the inferred variational distribution in latent space, the sample weights will be more smooth and have larger effective sample size [24] than the point-wisely estimated weights as previous literature [30] did.

To estimate $W_Z(\mathbf{X}, \mathbf{Z})$, we set the transformed data points from observational dataset $\{(\mathbf{x}_i, \mathbf{z})\}_{1 \leq i \leq n}$, $\mathbf{z} \sim q_\phi(\mathbf{z}|\mathbf{t}_i)$ as positive samples ($L = 1$) and data points from decorrelated target dataset $\{(\mathbf{x}_i, \mathbf{z})\}_{1 \leq i \leq n}$, $\mathbf{z} \sim p(\mathbf{z})$ as negative samples ($L = 0$). After fitting these data points into a deep neural networks based classifer $p_{\theta_d}(L|\mathbf{X}, \mathbf{Z})$, we can get the density ratio in the space $\mathcal{X} \times \mathcal{Z}$ via Bayes theorem:

$$W_Z(\mathbf{X}, \mathbf{Z}) = \frac{p(\mathbf{X}, \mathbf{Z}|L=0)}{p(\mathbf{X}, \mathbf{Z}|L=1)} = \frac{p(L=1)}{p(L=0)} \cdot \frac{p(L=0|\mathbf{X}, \mathbf{Z})}{p(L=1|\mathbf{X}, \mathbf{Z})} = \frac{p(L=0|\mathbf{X}, \mathbf{Z})}{p(L=1|\mathbf{X}, \mathbf{Z})}, \tag{5}$$

where the term $p(L|\mathbf{X}, \mathbf{Z})$ is estimated by the classifier $p_{\theta_d}(L|\mathbf{X}, \mathbf{Z})$, and the term $\frac{p(L=1)}{p(L=0)}$ equals one for all the data points. The diagram of our VSR algorithm is shown in Figure 2. The whole process above can be seen as a pre-processing of the training data, with which the confounding bias in the original data can be effectively reduced and any off-the-shelf predictive model can be seamlessly applied after that.

### 3.3 Outcome Prediction

Some work assume prior knowledge on the form of the outcome function [25]. But it may not be true in practice. Therefore, we apply deep neural network $f_{\theta_p} : \mathcal{X} \times \mathcal{T} \mapsto \mathbb{R}$ with model parameters $\theta_p$ to predict the outcome considering the potential complex relationship between outcomes and confounders/treatments. With the variational sample weights learned above, the prediction loss is defined as the empirical estimation of $\mathcal{E}_f^w$:

$$\mathcal{L}_{pre} = \frac{1}{n} \sum_{i=1}^n w_i^d \cdot \mathcal{L}(f_{\theta_p}(\mathbf{x}_i, \mathbf{t}_i), y_i), \tag{6}$$

where $\mathcal{L}(\cdot, \cdot)$ is loss function, such as square error for regression tasks, or cross entropy for classification tasks. With this prediction network $f_{\theta_p}$ trained on the re-weighted data which have less confounding bias, we can achieve accurate counterfactual prediction of outcomes.

## 4 Experiments

Evaluating the methods of counterfactual prediction problems requires the ground truth of different treatment outcomes, which can hardly be satisfied by the observational data in reality. To partially overcome this obstacle, we conduct experiments on both synthetic datasets and datasets from a simulator mimicking recommendation systems in real world to evaluate the effectiveness of our method.

### 4.1 Experiment Settings and Baselines

We compare our VSR algorithm with the following methods. They are all based on the same prediction network architecture for fair comparision.

- *DNN*: It directly uses deep neural networks to predict the outcomes given confounders and treatments without any pre-processing to the observational data.

- *DNN&$W_{raw}$*: Before training the deep predictive model, it assigns the sample weights calculated by point-wise density ratio estimation in the joint space of confounders and raw treatments.

- *DNN&$W_{AE}$*: It uses autoencoder to learn the low dimensional representations of treatments and assigns sample weights by point-wise density ratio estimation in the joint space of confounders and learned representations.

In counterfactual prediction problem, the common goal is to achieve lower counterfactual outcome prediction error of the treatments randomly assigned among the entire population regardless the confounders [11, 12]. Therefore, we need to evaluate the models in the unbiased testing dataset conforming to the distribution where the treatments are assigned independently of confounders. More specifically, we randomly shuffles the matches of confounders and treatments in the original biased dataset to create the unbiased testing dataset.

## 4.2 Synthetic Experiments

In this section, we give a brief overview of how to generate the synthetic datasets and demonstrate the effectiveness of our proposed method.

### 4.2.1 Dataset

To evaluate the effectiveness of different methods, we generate the synthetic datasets under different settings. We first generate the confounders $\mathbf{X} = (x_1, x_2, ...., x_d)$, where the elements follows independent normal distribution:

$$x_1, x_2, ..., x_d \overset{iid}{\sim} \mathcal{N}(0, 1).$$

In the observational dataset, for each sample, we assign the treatment $\mathbf{T} \in \mathcal{T} \subset \{0, 1\}^p$ based on confounder variables. Firstly, we compute

$$\mathbf{L} = \mathbf{X} \cdot \mathbf{A} + \varepsilon_L, \quad \mathbf{F} = \mathbf{L} \cdot \mathbf{B},$$

where $\mathbf{L} \in \mathbb{R}^k$ is the latent representation with dimension $k \ll p$, $\mathbf{A} \in \mathbb{R}^{d \times k}$, $\mathbf{F} \in \mathbb{R}^p$, $\mathbf{B} \in \mathbb{R}^{k \times p}$, $\varepsilon_L \in \mathbb{R}^k$ is a normal noise vector and $\mathbf{A}, \mathbf{B}$ are constant. Then, assuming $\{i_1, i_2, .., i_s\}$ are the $s$ bits with largest value in $\mathbf{F}$, each bit $t_j$ in the bundle treatment $\mathbf{T}$ is defined as

$$t_j = \begin{cases} 1 & j \in \{i_1, i_2, .., i_s\} \\ 0 & j \notin \{i_1, i_2, .., i_s\} \end{cases} \tag{7}$$

Since $\mathbf{A}$ and $\mathbf{B}$ are constant matrices, the treatment $\mathbf{T}$ is determined by the latent representation $\mathbf{L}$ of low dimension $k$. Due to the relationship between $\mathbf{L}$ and $\mathbf{X}$, the treatments and confounders are correlated.

The outcome is generated from a pre-defined function, determined by both confounders and treatments:

$$\mathbf{y} = \sum_{i=1}^{d} \sum_{j=1}^{p} x_i d_{i,j} t_j + \varepsilon_y, \tag{8}$$

where $\mathbf{D} = \{d_{i,j}\}_{1 \leq i,j \leq n}$ is a pre-defined matrix, and $\varepsilon_y$ is a normal noise. To evaluate different methods, we calculate the root mean square error (RMSE) of estimation in the testing dataset where the treatments are independent of the confounders.

In this experiment, we set the confounder dimension $d = 10$, latent dimension $k = 3$, the number of one-value bits in treatments $s = 5$, and the noise variable $\varepsilon_y \sim \mathcal{N}(0, 0.01^2)$.

| Setting 1:Fix sample size $n = 10000$, varying dimension of treatments $p$ | | | | | | | | |
|---|---|---|---|---|---|---|---|---|
| $p$ | $p = 10$ | | $p = 20$ | | $p = 30$ | | $p = 50$ | |
| Methods | Mean | STD | Mean | STD | Mean | STD | Mean | STD |
| DNN | 0.617 | 0.043 | 0.997 | 0.139 | 1.380 | 0.155 | 1.940 | 0.278 |
| DNN&W$_{\text{raw}}$ | 0.528 | 0.044 | 0.997 | **0.056** | 1.197 | 0.092 | 1.543 | **0.108** |
| DNN&W$_{\text{AE}}$ | 0.529 | 0.045 | 0.977 | 0.069 | 1.201 | 0.092 | 1.520 | 0.170 |
| DNN&W$_{\text{VSR}}$ | **0.476** | **0.037** | **0.946** | 0.067 | **1.126** | **0.085** | **1.506** | 0.152 |
| Setting 2:Fix dimension of treatments $p = 10$, varying sample size $n$ | | | | | | | | |
| $n$ | $n = 5000$ | | $n = 10000$ | | $n = 15000$ | | $n = 20000$ | |
| Methods | Mean | STD | Mean | STD | Mean | STD | Mean | STD |
| DNN | 0.677 | 0.083 | 0.617 | 0.043 | 0.658 | 0.159 | 0.434 | 0.063 |
| DNN&W$_{\text{raw}}$ | 0.647 | 0.073 | 0.528 | 0.044 | 0.631 | 0.160 | 0.385 | 0.075 |
| DNN&W$_{\text{AE}}$ | 0.624 | 0.063 | 0.529 | 0.045 | 0.589 | 0.072 | 0.400 | 0.066 |
| DNN&W$_{\text{VSR}}$ | **0.572** | **0.053** | **0.476** | **0.037** | **0.518** | **0.064** | **0.367** | **0.044** |

Table 1: The experiment results on synthetic datasets of different methods. Mean and STD refer to the average value and standard deviation of the RMSE results in independent experiments. The lower of these metrics, the better.

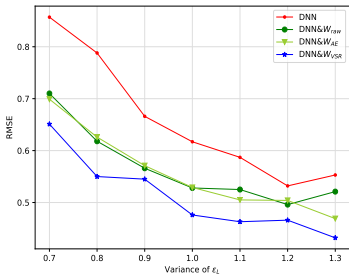

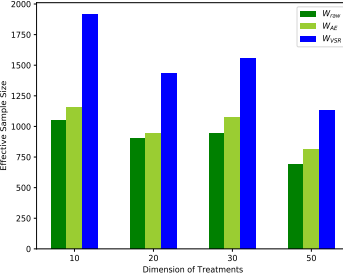

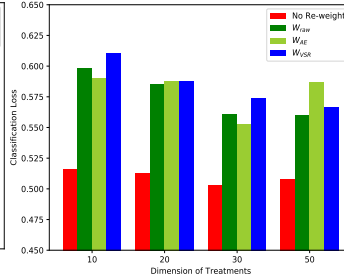

Figure 3: The testing RMSE when varying the variance of noise $\varepsilon_L$ in generating treatments.

Figure 4: The effective sample size of re-weighting methods, when fixing $n = 10000$ and varying $p$.

Figure 5: The degree of correlation (classification loss) comparision, when fixing $n = 10000$, varying $p$.

#### 4.2.2 Results

We conduct experiments under different settings by varying the sample size $n$ and the dimension of treatments $p$. For each experimental setting, we repeatedly carry out the experiments for 10 times and calculate the average value and standard deviation of RMSE in the testing dataset and report the prediction performance in Table 1.

From the results, we can clearly observe that compared with vanilla deep neural networks (DNN), training the predictive model on the re-weighted dataset can effectively reduce the prediction error of the model in the unbiased testing dataset. Owing to the decorrelation of treatments and confounders, sample re-weighting methods can significantly reduce the confounding bias in observational data and therefore produce more satisfactory results. Empirically, we further demonstrate the degree of correlation between the treatments and confounders before and after sample re-weighting, which can be measured by classification loss between the original pairs and shuffled pairs of confounders and treatments, that is $\{(\mathbf{x}_i, \mathbf{t}_i)\}_{1 \le i \le n}$ and $\{(\mathbf{x}_i, \mathbf{t}_{v_i})\}_{1 \le i \le n}$ where $\mathbf{v} = \{v_i\}_{1 \le i \le n}$ is a random permutation of sample index. It is intuitive that larger classification loss means less correlation between treatments and confounders. The results are presented in Figure 5. We can find that the correlation is significantly reduced after sample re-weighting. Among the re-weighting methods, W$_{\text{AE}}$ performs better than W$_{\text{raw}}$, since W$_{\text{AE}}$ also explores the latent structure of treatments and removes confounding bias better via decorrelating latent representations and confounders.

Our proposed re-weighting method $W_{VSR}$ achieves lower RMSE than re-weighting methods based on raw treatments $W_{raw}$ and representation from autoencoder $W_{AE}$, since our method learns the sample weights in a more smooth way through aggregating the density ratio in the variational distribution. Moreover, the smoothness of weights $\{w_i\}_{1 \le i \le n}$ can be measured by effective sample size [24] which is defined as $N_{eff} = \frac{(\sum_{i=1}^{n} w_i)^2}{\sum_{i=1}^{n} w_i^2}$. As shown in Figure 4, the effective sample size of our proposed $W_{VSR}$ is significantly larger than $W_{raw}$ and $W_{AE}$.

We also alter the variance of $\varepsilon_L$ in generating treatments while fixing the sample size $n = 10000$ and dimension of treatments $p = 10$. Intuitively, when $\varepsilon_L$ is smaller, the confounding bias in training is more severe and the prediction error in the unbiased testing dataset tend to be larger. The results are shown in Figure 3. Our method consistently achieves lowest prediction error across all settings.

### 4.3 Real World Experiments

Considering that few datasets contain ground truth of different treatment outcomes, we conduct experiments with simulator Recsim [13] which mimics the recommendation systems in real world.

#### 4.3.1 Data generation

There is a simulation environment about document recommendation [1] in Recsim. In this environment, a document $D_i$ is characterized by the topic (category) $c_i$ and quality $q_i$. A user is characterized by a vector of affinity to each document topic $\mathbf{X} \in \mathbb{R}^d$, where $d$ is the number of document topics. To generate observational dataset with confounding bias, we assigned the bundle treatment (i.e. recommended document list) by the following process:

- Intention vector of each document topic is generated as $\mathbf{L} = \mathbf{X} + \varepsilon_L, \varepsilon_L \sim \mathcal{N}(0, 0.81\mathbf{I})$.
- The recommending score of each document $D_i$ is calculated as the document quality plus the intention value of the document topic. Formally, $Score_i = l_{c_i} + q_i$.
- The $s$ documents with highest score are selected as recommended documents forming the bundle treatment.

Given the user features (confounder) and recommended documents (treatment), the simulator provide the ground truth of user's click rate. The methods is evaluated by the RMSE of click rate prediction in the unbiased testing dataset. We fixed the sample size $n = 10000$, the number of document topics $d = 4$ and selected documents $s = 4$. The user affinity and document quality is generated from standard normal distribution.

#### 4.3.2 Results

We vary the number of available documents and repeatedly conduct experiments 10 times for each setting. The results are reported in Table 2.

The overall results are quite consistent with the experiments on the synthetic datasets. Since sample re-weighting process can reduce the correlation between treatments and confounders in the observational data, the predictive model trained on this less biased pseudo-data can achieve more accurate prediction in the unbiased testing dataset. Our proposed variational sample re-weighting (VSR) algorithm enjoys a more smooth estimation of sample weights in the variational distribution space. Therefore, the prediction RMSE of our method can be smaller.

## 5 Conclusion

In this paper, we investigate the counterfactual outcome prediction for bundle treatment. We assume the bundle treatment has low dimensional latent structure and propose the variational sample re-weighting (VSR) algorithm to decorrelate the learned treatment representations and confounders. Via extensive experiments on synthetic datasets and real world datasets, we showed that the proposed sample weights can effectively reduce the confounding bias in the observational data and achieve more accurate counterfactual outcome estimation. For future works, we will attempt to learn the optimal treatment for each individual.

| RMSE of click rate prediction ($\times 10^{-2}$) | | | | | | | | |
|---|---|---|---|---|---|---|---|---|
| Document number $p$ | $p=10$ | | $p=20$ | | $p=30$ | | $p=50$ | |
| Methods | Mean | STD | Mean | STD | Mean | STD | Mean | STD |
| DNN | 2.694 | 0.589 | 3.941 | 0.716 | 4.415 | 0.582 | 4.443 | 0.613 |
| DNN&$W_{raw}$ | 1.950 | 0.517 | 3.258 | 0.621 | 3.856 | **0.455** | 3.788 | 0.625 |
| DNN&$W_{AE}$ | 1.711 | 0.407 | 3.312 | 0.741 | 3.683 | 0.515 | 3.623 | 0.619 |
| DNN&$W_{VSR}$ | **1.596** | **0.349** | **2.923** | **0.407** | **3.318** | 0.459 | **3.385** | **0.598** |

Table 2: Experiment results on real world datasets of different methods. Mean and STD refer to the average value and standard deviation of RMSE. Lower is better.

## Broader Impact

This work investigates in the problem of counterfactual outcome prediction. This can utilize the power of machine learning technology to assist better decision making in many domains. For example, the marketer can be helped to find the best marketing actions to improve user conversion. At the same time, this work may suffer from some risk. It relies on some standard assumption in causal inference field, such as unconfoundedness, stable unit treatment value. These assumptions may be violated in some scenarios. For example, some confounders, such as economic status, may not be measured due to ethical or technical reasons. The samples may also have interactions with each other in some scenarios. All these violation can bring risk to the prediction results and lead to poor decision making.

## Acknowledgments and Disclosure of Funding

This work was supported in part by National Key R&D Program of China (No. 2018AAA0102004), National Natural Science Foundation of China (Nos. U1936219, 61772304, 61531006, U1611461), Beijing Academy of Artificial Intelligence (BAAI), and a grant from the Institute for Guo Qiang, Tsinghua University. Bo Li's research was supported by the Tsinghua University Initiative Scientific Research Grant, No. 2019THZWJC11; National Natural Science Foundation of China, No. 71490723 and No. 71432004; Science Foundation of Ministry of Education of China, No. 16JJD630006.

## Footnotes

[1]https://github.com/google-research/recsim/blob/master/recsim/environments/interest_exploration.py

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
