[Supplementary Material]

# A Supplementary Material

## A.1 Proof

**Theorem 1.** *Assuming a family $G$ of functions $g : \mathcal{X} \times \mathcal{T} \mapsto \mathbb{R}$, $l(\mathbf{X}, \mathbf{T}) = \mathcal{L}(f(\mathbf{X}, \mathbf{T}), \mathbf{y}(\mathbf{X}, \mathbf{T})) \in G$ and overlap assumption is satisfied, formally, $p(\mathbf{T}|\mathbf{X}) > 0, \forall \mathbf{T} \in \mathcal{T}, \mathbf{X} \in \mathcal{X}$, we have:*

$$\mathcal{E}_{cf} \leq \mathcal{E}_f^w + \text{IPM}_G(W_T(\mathbf{X}, \mathbf{T})p(\mathbf{X}, \mathbf{T}), p(\mathbf{X})p(\mathbf{T})). \tag{1}$$

*More specifically, $\text{IPM}_G = 0$ and $\mathcal{E}_{cf} = \mathcal{E}_f^w$, when $W_T(\mathbf{X}, \mathbf{T}) = p(\mathbf{T})/p(\mathbf{T}|\mathbf{X})$.*

*Proof.* The inequality is equivalent to

$$
\begin{aligned}
\mathcal{E}_{cf} - \mathcal{E}_f^w &= \int_{\mathbf{X}} \int_{\mathbf{T}} (p(\mathbf{X})p(\mathbf{T}) - W_T(\mathbf{X}, \mathbf{T})p(\mathbf{X}, \mathbf{T}))\mathcal{L}(f(\mathbf{X}, \mathbf{T}), \mathbf{y}(\mathbf{X}, \mathbf{T}))d\mathbf{X}d\mathbf{T} \\
&\leq \left| \int_{\mathbf{X}} \int_{\mathbf{T}} (p(\mathbf{X})p(\mathbf{T}) - W_T(\mathbf{X}, \mathbf{T})p(\mathbf{X}, \mathbf{T}))\mathcal{L}(f(\mathbf{X}, \mathbf{T}), \mathbf{y}(\mathbf{X}, \mathbf{T}))d\mathbf{X}d\mathbf{T} \right| \\
&\leq \sup_{g \in G} \left| \int_{\mathbf{X}} \int_{\mathbf{T}} (p(\mathbf{X})p(\mathbf{T}) - W_T(\mathbf{X}, \mathbf{T})p(\mathbf{X}, \mathbf{T}))g(\mathbf{X}, \mathbf{T})d\mathbf{X}d\mathbf{T} \right| \\
&= \text{IPM}_G(W_T(\mathbf{X}, \mathbf{T})p(\mathbf{X}, \mathbf{T}), p(\mathbf{X})p(\mathbf{T})).
\end{aligned}
$$

When $W_T(\mathbf{X}, \mathbf{T}) = p(\mathbf{T})/p(\mathbf{T}|\mathbf{X})$,

$$
\begin{aligned}
&\text{IPM}_G(W_T(\mathbf{X}, \mathbf{T})p(\mathbf{X}, \mathbf{T}), p(\mathbf{X})p(\mathbf{T})) \\
&= \sup_{g \in G} \left| \int_{\mathbf{X}} \int_{\mathbf{T}} \left( \frac{p(\mathbf{T})}{p(\mathbf{T}|\mathbf{X})} p(\mathbf{X}, \mathbf{T}) - p(\mathbf{X})p(\mathbf{T}) \right) g(\mathbf{X}, \mathbf{T})d\mathbf{X}d\mathbf{T} \right| \\
&= \sup_{g \in G} \left| \int_{\mathbf{X}} \int_{\mathbf{T}} \left( \frac{p(\mathbf{T})p(\mathbf{X})}{p(\mathbf{X}, \mathbf{T})} p(\mathbf{X}, \mathbf{T}) - p(\mathbf{X})p(\mathbf{T}) \right) g(\mathbf{X}, \mathbf{T})d\mathbf{X}d\mathbf{T} \right| \\
&= 0
\end{aligned}
$$

$$
\begin{aligned}
\mathcal{E}_f^w &= \int_{\mathbf{X}} \int_{\mathbf{T}} \frac{p(\mathbf{T})}{p(\mathbf{T}|\mathbf{X})} p(\mathbf{X}, \mathbf{T})\mathcal{L}(f(\mathbf{X}, \mathbf{T}), \mathbf{y}(\mathbf{X}, \mathbf{T}))d\mathbf{X}d\mathbf{T} \\
&= \int_{\mathbf{X}} \int_{\mathbf{T}} \frac{p(\mathbf{T})p(\mathbf{X})}{p(\mathbf{X}, \mathbf{T})} p(\mathbf{X}, \mathbf{T})\mathcal{L}(f(\mathbf{X}, \mathbf{T}), \mathbf{y}(\mathbf{X}, \mathbf{T}))d\mathbf{X}d\mathbf{T} \\
&= \int_{\mathbf{X}} \int_{\mathbf{T}} p(\mathbf{T})p(\mathbf{X})\mathcal{L}(f(\mathbf{X}, \mathbf{T}), \mathbf{y}(\mathbf{X}, \mathbf{T}))d\mathbf{X}d\mathbf{T} \\
&= \mathcal{E}_{cf}
\end{aligned}
$$

$\square$

**Theorem 2.** *Defining $\hat{\theta}_p^{\mathbf{w}} = \arg\min_{\theta_p} \frac{1}{n} \sum_{i=1}^n w_i \cdot \mathcal{L}(f_{\theta_p}(\mathbf{x}_i, \mathbf{t}_i), y_i)$, counterfactual prediction error for model parameters $\theta_p$ as $\mathcal{E}_{cf}(\theta_p) = \mathbb{E}_{\mathbf{X} \sim p(\mathbf{X})}[\mathbb{E}_{\mathbf{T} \sim p(\mathbf{T})}[\mathcal{L}(f_{\theta_p}(\mathbf{X}, \mathbf{T}), \mathbf{y}(\mathbf{X}, \mathbf{T}))]]$ and assuming the error function $\mathcal{L}(f_{\theta_p}(\mathbf{X}, \mathbf{T}), \mathbf{y})$ is twice-differentiable and strictly convex on $\theta_p$, then*

$$\mathcal{E}_{cf}(\hat{\theta}_p^{\mathbf{w}^d}) = \inf_{\mathbf{w}} \mathcal{E}_{cf}(\hat{\theta}_p^{\mathbf{w}}), \tag{2}$$

*where $\mathbf{w}^d = \{p(\mathbf{t}_i)/p(\mathbf{t}_i|\mathbf{x}_i)\}_{i=1}^n$. This means $\mathbf{w}^d$ is one of the optimal sample weights that induce lowest counterfactual prediction error by minimizing prediction error on the re-weighted dataset.*

*Proof.* We define $\mathcal{E}_{cf}(\hat{\theta}_p^{\mathbf{w}^*}) = \inf_{\mathbf{w}} \mathcal{E}_{cf}(\hat{\theta}_p^{\mathbf{w}})$, $L(\mathbf{X}, \mathbf{T}, \theta_p) = \mathcal{L}(f_{\theta_p}(\mathbf{X}, \mathbf{T}), \mathbf{Y}(\mathbf{X}, \mathbf{T}))$ and $\hat{\theta}_p^{\varepsilon_j} = \arg\min_{\theta_p} \frac{1}{n} \sum_{i=1}^n w_i^* \cdot L(\mathbf{x}_i, \mathbf{t}_i, \theta_p) + \varepsilon_j \cdot L(\mathbf{x}_j, \mathbf{t}_j, \theta_p)$. It is obviously that

$$\left. \frac{\partial \mathcal{E}_{cf}(\hat{\theta}_p^{\varepsilon_j})}{\partial \varepsilon_j} \right|_{\varepsilon_j = 0} = 0, 1 \leq j \leq n \tag{3}$$

|                                              | Synthetic experiments | Real world experiments |
| -------------------------------------------- | :-------------------: | :--------------------: |
| Hidden units of layer for confounders        | 10                    | 4                      |
| Hidden units of layer for treatments          | 5                     | 5                      |
| Number of hidden layers in MLP                | 1                     | 1                      |
| Hidden units of MLP                           | 10                    | 10                     |
| Initial learning rate of prediction networks | 0.01                  | 0.01                   |
| Learning rate decay of prediction networks    | 0.98                  | 0.95                   |
| Epoch interval for each learning rate decay   | 5                     | 10                     |
| Epochs for training prediction networks       | 500                   | 100                    |
| Number of hidden layers in classifer          | 2                     | 2                      |
| Hidden units of classifer                      | 10                    | 10                     |
| Learning rate of classifer                     | 0.01                  | 0.01                   |
| Epochs for training classifer                  | 200                   | 200                    |

Table 1: The hyper-parameters in experiments.

According to results in [6, 7], we have

$$\frac{\partial \mathcal{E}_{cf}(\hat{\theta}_p^{\varepsilon_j})}{\partial \varepsilon_j}\bigg|_{\varepsilon_j=0} = -\mathbb{E}_{\mathbf{X}\sim p(\mathbf{X}),\mathbf{T}\sim p(\mathbf{T})}[\nabla_{\theta_p}L(\mathbf{X},\mathbf{T},\hat{\theta}_p^{\mathbf{w}^*})]H_{\hat{\theta}_p^{\mathbf{w}^*}}^{-1}\nabla_{\theta_p}L(\mathbf{x}_j,\mathbf{t}_j,\hat{\theta}_p^{\mathbf{w}^*}) = 0, 1 \le j \le n \tag{4}$$

where $H_{\hat{\theta}_p^{\mathbf{w}^*}} = \frac{1}{n}\sum_{i=1}^n \nabla_{\theta_p}^2 L(\mathbf{x}_i,\mathbf{t}_i,\hat{\theta}_p^{\mathbf{w}^*})$ is the Hessian matrix and positive definite (PD) when the risk is twice-differentiable and strictly convex on $\theta_p$. Therefore,

$$
\begin{aligned}
0 &= -\mathbb{E}_{\mathbf{X}\sim p(\mathbf{X}),\mathbf{T}\sim p(\mathbf{T})}[\nabla_{\theta_p}L(\mathbf{X},\mathbf{T},\hat{\theta}_p^{\mathbf{w}^*})]H_{\hat{\theta}_p^{\mathbf{w}^*}}^{-1}\sum_{j=1}^n w_j^d \nabla_{\theta_p}L(\mathbf{x}_j,\mathbf{t}_j,\hat{\theta}_p^{\mathbf{w}^*}) \\
&\approx -\mathbb{E}_{\mathbf{X}\sim p(\mathbf{X}),\mathbf{T}\sim p(\mathbf{T})}[\nabla_{\theta_p}L(\mathbf{X},\mathbf{T},\hat{\theta}_p^{\mathbf{w}^*})]H_{\hat{\theta}_p^{\mathbf{w}^*}}^{-1}\mathbb{E}_{\mathbf{X},\mathbf{T}\sim p(\mathbf{X},\mathbf{T})}[\frac{p(\mathbf{T})}{p(\mathbf{T}|\mathbf{X})}\nabla_{\theta_p}L(\mathbf{X},\mathbf{T},\hat{\theta}_p^{\mathbf{w}^*})] \\
&= -\mathbb{E}_{\mathbf{X}\sim p(\mathbf{X}),\mathbf{T}\sim p(\mathbf{T})}[\nabla_{\theta_p}L(\mathbf{X},\mathbf{T},\hat{\theta}_p^{\mathbf{w}^*})]H_{\hat{\theta}_p^{\mathbf{w}^*}}^{-1}\mathbb{E}_{\mathbf{X}\sim p(\mathbf{X}),\mathbf{T}\sim p(\mathbf{T})}[\nabla_{\theta_p}L(\mathbf{X},\mathbf{T},\hat{\theta}_p^{\mathbf{w}^*})]
\end{aligned}
$$

Since $H_{\hat{\theta}_p^{\mathbf{w}^*}}^{-1}$ is positive definite, $\mathbb{E}_{\mathbf{X}\sim p(\mathbf{X}),\mathbf{T}\sim p(\mathbf{T})}[\nabla_{\theta_p}L(\mathbf{X},\mathbf{T},\hat{\theta}_p^{\mathbf{w}^*})] = 0$. Because $L(\mathbf{X},\mathbf{T},\theta_p)$ is strictly convex on $\theta_p$ and $\mathbb{E}_{\mathbf{X}\sim p(\mathbf{X}),\mathbf{T}\sim p(\mathbf{T})}[\nabla_{\theta_p}L(\mathbf{X},\mathbf{T},\hat{\theta}_p^{\mathbf{w}^d})] \approx \sum_{j=1}^n w_j^d \nabla_{\theta_p}L(\mathbf{x}_j,\mathbf{t}_j,\hat{\theta}_p^{\mathbf{w}^d}) = 0$, it is proved that $\hat{\theta}_p^{\mathbf{w}^d} = \hat{\theta}_p^{\mathbf{w}^*}$. □

### A.2 Experiment details

The pseudo-code of VSR algorithm can be found in Algorithm 1.

The classifier for density ratio estimation is based on deep neural networks. Since excessively large neural networks may overfit the data points and produce extreme sample weights, we searched the number of hidden layers among 1 and 2, and the hidden units of each layer among 5 and 10 by grid searching.

Our algorithm and baselines are based on the identical outcome prediction network architecture. For fair comparasion, we ensure that the network complexity and the training process is identical for each method in the same experiment. Specifically, in the prediction networks, the treatments and confounders are firstly fed into two separate hidden layers. Then the output of the two hidden layers are concatenated and fed into a multi-layer perceptron (MLP) to predict the outcome. The model is trained using Adam [5], and we use the ELU [3] activation function. The hyper-parameters of the networks and training process are shown in Table 1.

In sythetic experiments, the matrices $\mathbf{A}$ and $\mathbf{B}$ are generated from independent gaussian distribution, that is $a_{i,j} \sim \mathcal{N}(0,1/9), b_{i,j} \sim \mathcal{N}(0,1)$. The matrix $\mathbf{D} = (3\mathbf{A}/4 + \mathbf{E}) \cdot \mathbf{B}$. $\mathbf{E}$ is generated from another gaussian distribution, that is $e_{i,j} \sim \mathcal{N}(0,0.25^2)$. In real-world dataset, the topic of each document is generated from uniform multinomial distribution.

---

**Algorithm 1** Variational sample re-weighting (VSR)

---
**Require:** Observational data $\{(\mathbf{x}_i, \mathbf{t}_i, y_i)\}_{1 \leq i \leq n}$, learning rate $\lambda_d$.

**Ensure:** Variational sample weights $\mathbf{w}^d = \{w_i^d\}_{i=1}^n$

1: Train VAE of treatment variables $\mathbf{T}$, including encoder $q_\phi(\mathbf{Z}|\mathbf{T})$ and decoder $p_\varphi(\mathbf{T}|\mathbf{Z})$.
2: Initialize the parameters $\theta_d$ of classifer $p_{\theta_d}(L|\mathbf{X}, \mathbf{Z})$.
3: **for** Batch $B = \{(\mathbf{x}_i, \mathbf{t}_i, y_i)\}_{i=1}^{|B|}$ in each iteration **do**
4:     Infer latent representations for each sample $\mathbf{z}_i^{pos} \sim q_\phi(\mathbf{z}|\mathbf{t}_i)$ and $\mathbf{z}_i^{neg} \sim \mathcal{N}(\mathbf{0}, \mathbf{I})$.
5:     Compute loss $\mathcal{L}_d^B = -\frac{1}{|B|}\sum_{i=1}^{|B|}[\log p_{\theta_d}(L=1|\mathbf{x}_i, \mathbf{z}_i^{pos}) + \log p_{\theta_d}(L=0|\mathbf{x}_i, \mathbf{z}_i^{neg})]$
6:     Update $\theta_d = \theta_d - \lambda_d \frac{\partial \mathcal{L}_d^B}{\partial \theta_d}$
7: **end for**
8: Define density ratio function $\mathbf{w}_Z(\mathbf{X}, \mathbf{Z}) = \frac{p_{\theta_d}(L=0|\mathbf{X}, \mathbf{Z})}{p_{\theta_d}(L=1|\mathbf{X}, \mathbf{Z})}$
9: Set $w_i^d = \frac{1}{\mathbb{E}_{\mathbf{z} \sim q_\phi(\mathbf{z}|\mathbf{t}_i)}[1/\mathbf{w}_Z(\mathbf{x}_i, \mathbf{z})]}$ for each sample.
10: **return** $\mathbf{w}^d = \{w_i^d\}_{i=1}^n$

---

## A.3 Deep Neural Networks with Independent Representation

We also bring the idea from domain adaptation [2, 4] to remove confounding bias. Considering the network architecture we employed, the prediction networks can be formed as $f_{\theta_p}(\mathbf{X}, \mathbf{T}) = h(\Phi_{\mathbf{X}}(\mathbf{X}), \Phi_{\mathbf{T}}(\mathbf{T}))$, where $\Phi_{\mathbf{X}}(\mathbf{X})$ and $\Phi_{\mathbf{T}}(\mathbf{T})$ is the output of hidden layers for confounders and treatments, and can be viewed as representations of confounders and treatments. This method removes confounding bias by constraining $\Phi_{\mathbf{X}}(\mathbf{X})$ and $\Phi_{\mathbf{T}}(\mathbf{T})$ are decorrelated. Therefore, the loss function consisting of prediction loss and correlation loss are defined as:

$$\mathcal{L}_{total} = \mathcal{L}_{pre} - \alpha \mathcal{L}_{cor}$$
$$\mathcal{L}_{pre} = \frac{1}{n}\sum_{i=1}^n \mathcal{L}(f_{\theta_p}(\mathbf{x}_i, \mathbf{t}_i), y_i),$$

where $\alpha$ is a hyper-parameter controlling the trade-off between outcome fitting and decorrelation constraint. Specifically, we set an exponentially increasing schedule $\alpha = 2/(1 + e^{-10\beta}) - 1$, as previous literature did [1, 4]. $\beta$ is the training progress linearly increasing from 0 to 1. In real world experiment, $\alpha$ is scaled down by 1000 times.

To measure the correlation between $\Phi_{\mathbf{X}}(\mathbf{X})$ and $\Phi_{\mathbf{T}}(\mathbf{T})$, we use a classifier $p_{\eta_d}(L|\Phi_{\mathbf{X}}(\mathbf{X}), \Phi_{\mathbf{T}}(\mathbf{T}))$ to distinguish the dataset $\{(\Phi_{\mathbf{X}}(\mathbf{x}_i), \Phi_{\mathbf{T}}(\mathbf{t}_i))\}_{1 \leq i \leq n}$ and $\{(\Phi_{\mathbf{X}}(\mathbf{x}_i), \Phi_{\mathbf{T}}(\mathbf{t}_{v_i}))\}_{1 \leq i \leq n}$ where $\mathbf{v} = \{v_i\}_{1 \leq i \leq n}$ is a random permutation of sample index. Larger classification loss means the distributions from which the two datasets are generated is closer and $\Phi_{\mathbf{X}}(\mathbf{X})$ is less correlated with $\Phi_{\mathbf{T}}(\mathbf{T})$. Therefore, $\mathcal{L}_{cor}$ is defined as

$$\mathcal{L}_{cor} = \min_{\eta_d} -\frac{1}{n}\sum_{i=1}^n [\log p_{\eta_d}(L=1|\Phi_{\mathbf{X}}(\mathbf{x}_i), \Phi_{\mathbf{T}}(\mathbf{t}_i)) + \log p_{\eta_d}(L=0|\Phi_{\mathbf{X}}(\mathbf{x}_i), \Phi_{\mathbf{T}}(\mathbf{t}_{v_i}))].$$

The prediction networks and classifier are simultaneously trained in adversarial using gradient reversal layer (GRL) [4].

We conduct the same experiments on synthetic datasets and real world datasets to evaluate the effectiveness of this methods. Figure 2 shows the degree of correlation before and after representation mapping. The degree of correlation is measured as the classification loss between the original pairs and shuffled pairs of confounders (representations) and treatments (representations). We can observe that the confounding bias is significantly reduced after representation mapping. As shown in the evaluation results of different experiments, the performance of this methods is worse than our proposed VSR method.

| Setting 1:Fix sample size $n = 10000$, varying dimension of treatments $p$ | | | | | | | | |
|---|---|---|---|---|---|---|---|---|
| $p$ | $p = 10$ | | $p = 20$ | | $p = 30$ | | $p = 50$ | |
| Methods | Mean | STD | Mean | STD | Mean | STD | Mean | STD |
| DNN&IR | 0.624 | 0.059 | 1.059 | 0.118 | 1.377 | 0.164 | 1.930 | 0.302 |
| Setting 2:Fix dimension of treatments $p = 10$, varying sample size $n$ | | | | | | | | |
| $n$ | $n = 5000$ | | $n = 10000$ | | $n = 15000$ | | $n = 20000$ | |
| Methods | Mean | STD | Mean | STD | Mean | STD | Mean | STD |
| DNN&IR | 0.667 | 0.119 | 0.624 | 0.059 | 0.639 | 0.096 | 0.435 | 0.068 |

Table 2: The experiment results on synthetic datasets of DNN&IR. Mean and STD refer to the average value and standard deviation of the RMSE results in independent experiments. The lower of these metrics, the better.

Figure 1: The testing RMSE of DNN&IR when varying the variance of noise $\varepsilon_L$ in generating treatments.

Figure 2: The degree of confounding bias (classification loss) with/without independence constraints

| RMSE of click rate prediction ($\times 10^{-2}$) | | | | | | | | |
|---|---|---|---|---|---|---|---|---|
| Document number $p$ | $p = 10$ | | $p = 20$ | | $p = 30$ | | $p = 50$ | |
| Methods | Mean | STD | Mean | STD | Mean | STD | Mean | STD |
| DNN&IR | 3.032 | 0.766 | 3.954 | 0.803 | 4.663 | 0.697 | 4.600 | 0.620 |

Table 3: Experiment results on real world datasets of DNN&IR. Mean and STD refer to the average value and standard deviation of RMSE. Lower is better.

## A.4  Results of synthetic experiments for non-constant s

To demonstrate that our method still works well when the number of one-value bits in treatment is not fixed, we repeat the synthetic experiments. In this experiments, **T** are generated as following:

$$t_j = \begin{cases} 1 & f_j > g \\ 0 & f_j \le g \end{cases} \tag{5}$$

where $g = \frac{p}{10} - 1$. The results are reported in Table 4.

## A.5  Results of synthetic experiments under misspecification of the dimension of latent space

The dimension of latent factors is hardly known in many scenarios. We repeat the synthetic experiments varying the dimension of latent space of VAE while the fixing sample size $n = 10000$ and variance of $\varepsilon_L$ is 1.0. As shown in Figure. 3, our method is not affected much by the misspecification of the dimension of latent space.

| Fix sample size $n = 10000$, varying dimension of treatments $p$, non-constant $s$ | | | | | | | | |
|---|---|---|---|---|---|---|---|---|
| $p$ | $p = 10$ | | $p = 20$ | | $p = 30$ | | $p = 50$ | |
| Methods | Mean | STD | Mean | STD | Mean | STD | Mean | STD |
| DNN | 0.884 | 0.121 | 1.126 | 0.097 | 1.828 | 0.130 | 3.244 | 0.447 |
| DNN&W$_\text{raw}$ | 0.778 | 0.081 | 1.100 | 0.078 | 1.712 | **0.117** | 3.143 | 0.288 |
| DNN&W$_\text{AE}$ | 0.775 | 0.084 | 1.125 | **0.064** | 1.618 | 0.121 | 3.108 | 0.271 |
| DNN&W$_\text{VSR}$ | **0.657** | **0.068** | **0.975** | 0.086 | **1.418** | 0.125 | **2.898** | **0.211** |
| DNN&IR | 0.893 | 0.107 | 1.131 | 0.136 | 1.819 | 0.145 | 3.200 | 0.273 |

Table 4: Experiment results for non-constant s on synthetic datasets. Mean and STD refer to the average value and standard deviation of the RMSE results. The lower of these metrics, the better.

(a) $p = 10$      (b) $p = 20$      (c) $p = 30$      (d) $p = 50$

Figure 3: The testing RMSE of DNN&W$_\text{VSR}$ when varying the dimension of latent space of VAE $k'$. The true dimension of latent space $k = 3$.