[Reviews · NeurIPS 2020]

Review 1

Summary and Contributions: The authors propose a method for doing weighted sample adjustment for learning counterfactual regression models when treatments are high-dimensional. The method uses a VAE to learn a low-dimensional representation of the treatments, and then finds weights that balance these representations. The authors demonstrate that the method works well in a synthetic example and a recommender system experiment.

Strengths: The paper lays out the problem quite clearly, and explains the method well. The experiments are well-designed, and show that each element of the method adds value over a simpler baseline.

Weaknesses: The framing of the problem is a little bit off. The issue here is not confounding bias, but rather variance reduction. This is not fatal for the paper, and actually suggests that the method can be widely applied under weak assumptions.

Correctness: Broadly, the claims are correct, although there is a minor issue with what is meant by "confounding bias" (see longer comments).

Clarity: Paper is clearly written.

Relation to Prior Work: Broadly, previous work is well-cited. A couple of additional citations: For methods that weight representations: Johansson, Kallus, Shalit, and Sontag https://arxiv.org/abs/1802.08598, but this is in the binary treatment setting. The strategy for estimating weights W_Z is the permutation weighting method from Arbour, Dimmery, and Sondhi https://arxiv.org/abs/1901.01230, so this is worth citing.

Reproducibility: Yes

Additional Feedback: As mentioned above, I think this method is very nice, but should be framed differently. In particular, the issue being addressed is not confounding _bias_; it is sample inefficiency when estimating the regression model f_{\theta_p}. This distinction is important in the causal inference literature, because a bias does not disappear with sample size. However, in this context, under the unconfoundedness assumption, if the model f_{\theta_p} is sufficiently flexible, it will converge to the same true counterfactual model in the large sample limit regardless of how the data are weighted (this is consistent with the experiments in the paper). In other words, the population risks E_{cf} and E_f^w are minimized at the same function. This is a direct consequence of the unconfoundedness assumption, and is the basis for several unweighted outcome regression approaches to causal inference (several of these are reviewed in Kunzel et al https://arxiv.org/abs/1706.03461). The irrelevance of weighting for bias in the context of using flexible outcome models has also been noted in Byrd and Lipton https://arxiv.org/abs/1812.03372. However, even though there is no confounding bias, weighting can help to reduce the variance of counterfactual estimates. In particular, up-weighting examples that are more common in the fully balanced case where p_bal(X, T) = p(X)p(T) than they are in the observational distribution p_emp(X, T) can cause the model f_{\theta_p} to focus more of its capacity on getting the outcome prediction for these examples right, rather than focusing on the most common combinations. This is discussed in the introduction, and perhaps this is what the authors mean by “confounding bias”. If this is the case, I would simply suggest that the authors use a different word. One nice consequence of this fact is that the method should be robust to misspecification of the VAE model. For example, if the dimension of the latent space is misspecified, or if the encoder has very restricted capacity and can’t capture the true manifold that the treatments lie close to, this method should still work, so long as the weights remain finite. Likewise, if smoothing heuristics are applied to the weights in the case where the variance is high, this would also not induce a bias. It might be useful to perform an experiment or two showing this robustness to misspecification.


Review 2

Summary and Contributions: This paper provides a method for causal inference when treatments are of bundle type (e.g., an advertisement banner with multiple products on sale). Such treatments are represented as high-dimensional, yet sparse, multiple binary vectors of form {0,1}^p where p is the number of products in the advertisement pool. The proposed method learns a low dimensional representation of the bundle treatments which is later used to derive variational sample weights in order to account for selection bias. The central contribution of this work appears to be Theorem 1, which is wrong (please see section on “correctness” for more details).

Strengths: + The empirical evaluations seem sufficient. + Causal inference is definitely a trending interest to the NeurIPS community.

Weaknesses: + I find the significance novelty of this work to be minimal; since the “latent treatment representation learning” part is trivial and the “variational sample weight learning” part is already published in [6, 23]. The central contribution of this work appears to be Theorem 1; however, the proof seems to be wrong. + In the supplementary material, the authors introduced an updated version of the objective function that they actually use in their experiments. This makes reproducibility impossible for people who only have access to the main paper. The main paper should include all the vital information on reproducing the work while the supplementary material should only be used for examples, or reminding the readers of already known results.

Correctness: Theorem 1 is wrong since the integral probability metric can only be defined on probability measures; however, W_T(X;T)p(X;T) and p(X)p(T) are not probability measures.

Clarity: Yes, the paper is well-written and easy to understand.

Relation to Prior Work: The paper provides an overview of the causal inference works with binary treatments and multi-level treatments, as well as sample re-weighting methods. However, there is neither any review of the literature on bundle treatments, nor any discussion on how this work differs from previous contributions.

Reproducibility: No

Additional Feedback: Lines 36 and 66: Ref. [10] proposed sample re-weighting “on top of” (as opposed to “instead of”) the representation learning approach in [25]. -------------------------------------- Post rebuttal comment: The authors' feedback have addressed my major concerns. I have modified my evaluation accordingly.


Review 3

Summary and Contributions: This work proposes a novel variational sample re-weighting method to eliminate confounding bias by decorrelating the treatments and confounders. The underlying assumption is that there exists a low dimensional latent structure which could generate the original treatment. Therefore, the proposed framework proceeds with the confounding bias reduction through inferring the latent factors of treatments and decorrelating them with confounders. Validity of the proposed method is shown through various synthetic and real datasets.

Strengths: 1. The presented method can serve as a pre-processing step before any predictive algorithm to reduce the effect of confounding bias. 2. Variational autoencoders (VAE) are leveraged for latent treatment representation learning in order to avoid strong assumption on the data generation process and latent structure. 3. A theoretical upper bound is provided for the average prediction error E_cf. 4. Simulation results show higher performance compare to other methods. Paper is well-written and easy to follow.

Weaknesses: 1. For a more rigorous performance evaluation, it would have been better to compare the proposed method with those of multi-level treatment or naïve extensions of single treatment settings. 2. Although the proposed framework lead to better results, as shown in Tables 1 and 2, the improvement is very marginal.

Correctness: Yes

Clarity: Yes

Relation to Prior Work: Yes

Reproducibility: Yes

Additional Feedback:


Review 4

Summary and Contributions: The paper proposes a method for predicting counterfactual outcomes in cases in which there are multiple treatments (and any subset of them can be chosen). They assume there is a latent, lower-dimensional representation of treatments which can be used alongside a variational approach for reweighting samples at prediction time. They then show that their method outperforms several ablations on a series of simulations.

Strengths: I am not incredibly well-versed in the literature, but to my knowledge, there are not methods specifically made for the bundle treatment scenario. From this standpoint, this work feels novel, and I could imagine it spurring future work and thought on this question.

Weaknesses: The weaknesses to me primarily arise from the limitations of the experiments. While I recognize that it's hard to find an actual real-world dataset in this area, it would be interesting if the experiments were a little more "adventurous" as a result, e.g. testing what happens when the strict assumptions made (e.g. of the latent dimensionality) are broken and variables are differed, since anything in real life is unlikely to follow the exact same generating process. While other multi-treatment algorithms are likely not optimized for this setting, it would still be interesting to include them in comparison.

Correctness: What I read of the methods appeared correct, but I did not read the proof of Theorem 1 in the Supplemental Material. The empirical methodology appears correct as well, but as I point out, would benefit from more stress testing.

Clarity: The paper is clearly written and easy to understand, though as I note later there are a few points that are left ambiguous that would be useful to clarify, e.g. how certain matrices were created. In addition, it would be good if notation used in Figure 2 (e.g. Z_neg) was also used in the text (or vice versa.)

Relation to Prior Work: Yes, the related work section makes it clear how this work differs from much of the related work, but it does not compare to any of the other models mentioned. It discusses how other models could deal with bundle treatments by moving into a multi-treatment setting with binary encoding, but it never compares to any of these methods. While such methods may not be feasible with p=50, it would be interesting to see how they compare to smaller p. I believe some related work is missing, which I expound upon below.

Reproducibility: No

Additional Feedback: - In terms of reproducibility, it would be useful if the authors provided their simulated and Recsim datasets, so that other methods and future experiments can be run on them. As a whole, the experiments were described in a reproducible manner, but there were details (e.g. how are A, B generated) that were left vague. (Code obviously helps as well.) It's unclear how Recsim is generating its dataset. - Was slightly misleading to call the RecSim dataset a real-world dataset - As far as I understand it, the training and test set arise from the same distribution, so it's unclear to me what performance would be like if treatment were assigned randomly in the test set, as opposed to aligning with the known confounders. - As mentioned earlier, at least the simulated experiment (and perhaps also the real-world) is based solely on additive treatment effect. While this assumption may hold in certain scenarios (e.g. click rates), it may not hold in fields like healthcare, and while I think that's a totally reasonable assumption/simplification to make, I think there should either be (i) experiments with non-additive effects, or (ii) it should be listed as a potential limitation. - In each experiments, 's' is selected to be a constant, so we never see a bundle with a different number of treatments/recommendations. Given the additive model used to generate the simulated data set, it would be interesting to see how the model would perform if 's' was also allowed to be a random variable. - Since it is unlikely we know the true underlying k in real life, it would be interesting to know performance when the true k doesn't match to understand what performance may look like in the real world. - Perhaps I'm misunderstanding the recommendation setup, but if we were to know which documents were clicked on by the user, it seems we could treat it as a simpler prediction problem than a bundle treatment problem. - I also feel some related work in the multi-treatment space is missing. For example, it would be good to consider or mention additive effects models (e.g. "Calculating additive treatment effects from multiple randomized trials provides useful estimates of combination therapies"), "Contextual Multi-Armed Bandits", "Causal Inference With General Treatment Regimes: Generalizing the Propensity Score", as well as some of the ideas covered in "Estimation of causal effects with multiple treatments: a review and new ideas." - Perhaps a quick spellcheck would be useful for a few typos, e.g. prediction and double periods in line 172 ______________________________________ Increasing my score in response to author's thorough rebuttal of most of my questions above

[Author Response · NeurIPS 2020]

We thank the reviewers for the valuable suggestions and appreciate the positive feedback.

**To Reviewer #1**:

**Framing of problem**. We agree that our method indeed improves the effective training sample size and reduces the
variance of counterfactual estimates. In this paper, we refer confounding bias to the correlation between $\mathbf{X}$ and $\mathbf{T}$ in the
training data which causes the distribution shift of the test distribution where $\mathbf{T}$ is randomly assigned. We will further
improve the problem framing in new version..

**To Reviewer #2**:

**Correctness of Theorem 1.** There might be some misunderstanding here. $P_W(\mathbf{X}, \mathbf{T}) = W_T(\mathbf{X}, \mathbf{T})P(\mathbf{X}, \mathbf{T})$ is defined
to be a valid distribution probability density if $\mathbb{E}_{\mathbf{X}, \mathbf{T} \sim P(\mathbf{X}, \mathbf{T})}[W_T(\mathbf{X}, \mathbf{T})] = 1$. Our sample weights $W_T(\mathbf{X}, \mathbf{T}) =$
$\frac{P(\mathbf{T})}{P(\mathbf{T}|\mathbf{X})}$ shown in line 118 can guarantee this property. And in the Lines 100, 102, our definition of countefactual
risk $\mathcal{E}_{cf}$ implies that our testing distribution is $P_{cf}(\mathbf{X}, \mathbf{T}) = P(\mathbf{X})P(\mathbf{T})$, which is also a valid probability density.
Therefore, the integral probability metric is defined on valid distributions.

**Novelty of our method.** We double-checked the literature [6,23], finding no technical content on variational sample
weight learning in these works. As far as we know, we are the first to explore the low dimensional latent structure of
bundle treatments and variational sample reweighting is proposed to address the challenge of decorrelating X and T.

**Is the objective function in supplementary material a updated version?** The objective function in the supplementary
material is not the updated version of our method. It is a baseline for comparison.

**To Reviewer #3**:

**Comparison to extension of methods in single treatment settings.** Many conventional methods of multi-level or
binary treatment build prediction model for each treatment and balance the confounder distribution of each treatment
group. In the bundle treatment setting, the number of groups can be large and the number of samples in each group can
be few. There may even be such cases that when we predict outcome of some treatment $\mathbf{t}$ in test set, there are no training
sample in the corresponding group. Therefore, the data insufficiency problem make these methods infeasible. There
are some conventional methods that view treatments as input features to make prediction such as BNN in "Learning
representations for counterfactual inference". The baselines DNN&$W_{\text{raw}}$ and DNN&IR can be viewed as extension of
these conventional methods incorporating the sample re-weighting and treatment invariant representation learning.

**To Reviewer #4**:

**Details of datasets** The matrices $A$ and $B$ are generated from standard gaussian distribution, that is $a_{i,j}, b_{i,j} \sim \mathcal{N}(0, 1)$.
We will refine the details and release the code and dataset in the future.

**Are Distribution of training and test set the same?** Since we randomly shuffled the matches of confounders and
treatments, the treatments are randomly assigned in test set. Meanwhile, the confounders and treatments are correlated
in training distribution. Therefore, the distribution of training set and test set are not the same.

**Are our method limited to additive treatment effect?** We check the outcome generation in Recsim dataset is of the
form $y = 1 - \frac{1}{1+\sum_i P_i}$, where $P_i$ is determined by the attributes of user and $i^{th}$ document. This does not belong to
additive treatment effect. Also, our method is designed without requirement for additive treatment effect model.

**Performance when s is not constant.** We define $t_j = 1$ if $f_j > e$, otherwise $t_j = 0$. Then the number of treatments is
not constant. We set threshold $e = p/10 - 1$. See results in Table 1.

**Performance when some information is missing.** We conduct experiments when the dimension of latent representa-
tion $k'$ in VAE is mismatched with true $k$. The results in Table 1 shows that our method is robust to this mismatch, even
when $k'$ is smaller than true $k$, demonstrating the effectiveness of our method in the case of missing information.

**Whether we know which document is clicked.** In our setup, we do not know which document is clicked. The
probability of clicking bundle is simultaneously determined by all the documents in bundle and the user.

| Fix sample size $n = 10000$, varying dimension of treatments $p$, non-constant $s$ | | | | | | | |
|---|---|---|---|---|---|---|---|
| $p$ | $p = 10$ | | $p = 20$ | | $p = 30$ | | $p = 50$ | |
| Methods | Mean | STD | Mean | STD | Mean | STD | Mean | STD |
| DNN | 0.884 | 0.121 | 1.126 | 0.097 | 1.828 | 0.130 | 3.244 | 0.447 |
| DNN&$W_{\text{raw}}$ | 0.778 | 0.081 | 1.100 | 0.078 | 1.712 | **0.117** | 3.143 | 0.288 |
| DNN&$W_{\text{AE}}$ | 0.775 | 0.084 | 1.125 | **0.064** | 1.618 | 0.121 | 3.108 | 0.271 |
| DNN&$W_{\text{VSR}}$ | **0.657** | **0.068** | **0.975** | 0.086 | **1.418** | 0.125 | **2.898** | **0.211** |
| DNN&IR | 0.893 | 0.107 | 1.131 | 0.136 | 1.819 | 0.145 | 3.200 | 0.273 |

| Fix sample size $n = 10000$, dimension of treatments $p = 10$, varying $k'$ | | | | | | | |
|---|---|---|---|---|---|---|---|
| $k'$ | $k' = 2$ | | $k' = 3 = k$ | | $k' = 4$ | | $k' = 5$ | |
| Methods | Mean | STD | Mean | STD | Mean | STD | Mean | STD |
| DNN&$W_{\text{VSR}}$ | 0.506 | 0.051 | 0.476 | 0.037 | 0.479 | 0.032 | 0.484 | 0.050 |

Table 1: Experiment results for non-constant s and varying $k'$.

[Meta-Review · NeurIPS 2020]

The authors propose a method for doing weighted sample adjustment for learning counterfactual regression models when treatments are high-dimensional. The reviewers, after some discussion, converged on the view that the paper is a nice contribution to the estimation theory for causal effects. One area where the paper could benefit is a discussion of the connections of the author's results to results on semi-parametric efficiency theory and influence functions (see e.g. comments of Reviewer 1). It is likely there is a close relationship between the role weights play in improving efficiency and efficient influence functions for the problem (even under randomization).